# Family care and predictors of the disabled elderly in China: A cross-sectional study based on the Anderson model

Yuwei Zhang, Li Wang *

School of Health Services Management, Southern Medical University, Guangzhou, China

* trmd_2002@163.com

## Abstract

In light of China's progressively aging population, family care with the "warmth of affection" has always been an irreplaceable form of care that meets the wishes of the majority of disabled elderly to enjoy their twilight years comfortably. Data from a follow-up survey on the influencing factors of the Chinese Longitudinal Healthy Longevity Survey from 2018 and the Anderson Model was used as a theoretical framework to analyze the influencing factors using a binary logistic regression model. Children were the main providers of care services among disabled elderly in family care. Family care for disabled elderly was influenced by the combined effects of age ($P < 0.01$), gender ($P < 0.05$), marital status ($P < 0.01$), number of children ($P < 0.01$), housing ownership ($P < 0.05$), primary carer preference ($P < 0.05$), and self-assessed health ($P < 0.01$) were jointly affected. As a result, it's critical to set up a strong social support network that is focused on family caregiving and includes tailored interventions based on the requirements of disabled elderly and family caregivers.

**Data Availability Statement:** Data are available in a public, open access repository. The datasets used and/or analysed during the current study are available from the https://opendata.pku.edu.cn/dataset.xhtml?persistentId=doi:10.18170/DVN/

## Introduction

Socio-demographic ageing is a long-term trend in global demographic change, and as life expectancy increases, the number of impaired older people grows. The large number of disabled elderly has had varying degrees of impact on society and families, and although the problem of caring for disabled elderly caused by aging requires an active response from society as a whole, However, due to external resource restrictions, the national old-age security system is not yet strong enough to meet the current need for care, and the bulk of disabled elderly turn to family care. Studies have shown that more than 80% of older people worldwide want to be cared for by their children [1], and in the United States, more than half of older people aged 65 years or older prefer family care models, in practice, 40–60% provide care services to family members or relatives and friends, and nearly 30% provide care support at least once a week [2], with family care being the main choice for older people with disabilities [3], although the rapid development of Western countries after the Second World War led to better security for the elderly and problems arising from institutional care, family care remained a key issue in gerontology until the 1970s [4]. The issue of family care for disabled elderly is a global concern

XRV2WN. The database is accessible to researchers.

**Funding:** This study was funded by A Study on Guangzhou's Active Response to Population Ageing: Identification of Dilemmas and Breakthroughs in Guangzhou's Smart Health and Aging Services under the Ecosystem View (2022GZYB37), A Study on the Holistic Governance of the Multiple Supply of Long-term Care Services in China(20FGLB048), Guangdong Social Science Research Base "Health Policy and Health Governance Research Center". The funders had no role in study design, data collection and analysis, decision to publish, or preparation of the manuscript.

**Competing interests:** The authors have declared that no competing interests exist.

**Abbreviations: CLHLS**, Chinese Longitudinal Longevity Survey (CLHLS).

as well as a societal issue confronting China throughout its present demographic shift. China presently boasts the world's greatest elderly population, and by the end of 2022, China will have 280 million elderly individuals aged 60 years and older, 44 million of them will be disabled, with the vast majority choosing to live at home and be cared for by family members. On the one hand, family care is necessary in long-term care due to emotional attachment and family duties. On the other hand, the traditional Chinese notion of upbringing is so strongly ingrained that family care is frequently the first choice for the majority of the disabled elderly. According to studies, the utilization rate of family care exceeds 90%, with spouses or children of disabled elderly serving as the most common caregivers, and that elderly people with self-care impairment prefer family care over those who can live a self-care life, and that the trend of family care for disabled elderly in China is increasing with the development of society and the passage of time [5–9].

The family is the fundamental social unit that keeps society working normally, and it plays a significant role in the development of an all-encompassing, scientific long-term care system. Family caregivers now have to manage several social tasks due to the nuclearization of the family and the empty nest phenomenon in society. Family caregivers are obligated to shoulder a heavy physical and mental burden due to the demands of both their caregivers and their families. giving family care entails giving long-term, labor-intensive, and energy-intensive care. According to similar studies, providing long-term care can increase a caregiver's risk of developing anxiety, despair, and physical discomfort [10]. Furthermore, the physiological, psychological, and behavioral traits of the disabled elderly, along with the caregivers' own health status, social background, caregiving hours, social support, and other factors, all influence the prevalence of caregiving burden among these caregivers [11]. Some researchers have noted that there is a negative correlation between the quality of family care and the burden of care, and that many caregivers experience stress from both social and familial demands on top of their already heavy caregiving responsibilities, which can result in a deterioration in the quality of family care [12]. Inadequate management family conflicts are likely to get worse when family care is provided in conjunction with social work, which can have an adverse effect on care quality and create problems for the long-term viability of family care.

Many studies on family caregiving for disabled elderly are currently available. Studies on family caregiving for disabled elderly have been conducted for a longer time in developed Western countries. According to some academics, financial support for disabled elderly can increase their motivation to care for their family members and make them more likely to take the initiative to do so [13]. "The National Long-Term Care Survey (NLTCS) indicates that family income influences whether or not a disabled elderly receives family caregivers. It has also been discovered that poor levels of education among the disabled elderly result in fewer caregivers with greater unmet needs. A review of pertinent empirical studies reveals that economic income, health status, and family resources all have a significant impact on the extent to which the disabled elderly' family care needs are met [14]. An analysis of pertinent empirical research demonstrates that family resources, health status, and economic income all have a significant impact on how well the requirements of disabled elderly are met by family caregivers [15]. The Chinese study emphasizes the geographical aspect of the research in addition to concentrating on the individual and family characteristics of the disabled elderly. The results indicate that there are notable differences between urban and rural locations in terms of the unmet demands of family care for the disabled elderly. Therefore, raising the standard of family care services requires a thorough awareness of the state of family care for disabled elderly well as an analysis of the predicting indicators.

Focusing on the current state and determinants of family caring for the disabled elderly is crucial because of the caregiving advantage that comes with kinship and a wide range of

support networks. Meanwhile, in terms of research paradigm, relevant studies have made limited use of theoretical frameworks and models, there is a lack of empirical studies of professional survey data, research on pathological problems of disabled elderly and a comprehensive and systematic summary of the elements influencing family care for the disabled elderly. This study is based on The Andersen Model, also known as The Behavioral Model of Health Services Use (BMHSU), which Dr. Andersen developed in 1986. It was first developed to analyze health-care behaviors, and the 2013 version is the main model for research in this area, with Environment, Population Characteristics, Health Behavior, and Outcomes as the research framework [16, 17]. In an attempt to analyze the long-term care demands of the elderly and the factors influencing them in China's middle and upper sections of the Yangtze River, Zeng et al. added psychological components to the Anderson model. Kempen et al. conducted a study that combined the Anderson model with the use of professional family care by the elderly, and Yong et al. examined the current status of the elderly's preference for the combination of healthcare and pensions, as well as the factors influencing their use of the combination of healthcare and pensions in long-term care facilities based on the Anderson model [18–20].

Family care for disabled elderly is essentially the utilisation of family care services, which is common to the utilisation of health services. As a result, this study employs Anderson's model as the framework for analyzing the 2018 data from CLHLS, with the goal of evaluating family care for disabled elderly and its related influencing factors, compensating for a lack of relevant empirical evidence, and bringing into play the key role of family care, in order to provide empirical references for the improvement of family care policy and optimisation of the system.

## Materials and methods

### Data source and study population

This study uses data from the 2018 Tracking Survey on Factors Influencing the Health of the Elderly in China, which was carried out in collaboration with the Peking University and Duke University Center for Healthy Aging Development Research. The survey included 15,874 elderly individuals over 65 who lived in 23 different Chinese provinces, cities, and autonomous regions. The extensive database coverage, the survey's scientific format, and the evenly distributed senior population in cities and rural areas all contribute to this study's increased dependability [21].

Disabled elderly are the focus of this study, and the daily activity ability in the questionnaire is compared to the international standard Katz scale to choose six daily activities (eating, dressing, bathing, going to the bathroom, controlling urination and defecation, indoor activities), with at least one item indicating that disabled elderly require assistance from others [22]. Anomalies and missing values of the key indexes were then removed, eliminating the sample of 1,318 people.

### Variables and measures

### Dependent variable

The person providing family care is the dependent variable. Since children and spouses provide the majority of family care, this paper only examines the care given by spouses and children, where "children" refers to both the children of disabled elderly and the daughters- and son-in-laws who live with them [23]. "When you need help from others in 6 activities of daily living, who is the main helper?" asks the quiz. and "Whom do you now reside with? and "Who do you currently live with?" were chosen, and the care given by spouses and children in the home environment was given a value of 1, while the care given by other relatives and friends—

aside from spouses and children—was given a value of 0. Disabled elderly who have a caregiver and receive informal care mostly at home were selected.

### Independent variables

The factors of housing ownership, the willingness of the primary caregiver, and the availability of community social services were added in accordance with Chinese national conditions, and the indicators were categorized into three categories based on Anderson Model: predisposing factors, enabling factors, and need factors. The value of "no difficulty" was allocated as one point in each of the six areas: eating, bathing, dressing, toileting, peeing and defecating, and indoor activities. The values of "some difficulty" and "unable to complete" were combined to assign two points each. A total score of 7–8 is thought to be moderate. Mild disability is defined as a total score of 7–8, moderate disability as 9–10, and severe disability as 11–12 [24]. According to the questionnaire, "community living care" includes "living care and daily shopping"; "community medical service" includes "home delivery of medicine and doctor visits"; and "community medical service" includes "community medical service." "Community medical services" [5, 25].

### Statistical analysis

Using the two-dimensional indicators of "whether the spouse provides care" and "whether the children provide care," this study empirically analyzes the antecedent, enabling, and need factors affecting family caregiving for disabled elderly. The family caregiver is the dependent variable, and regression analysis uses a binary logistic model with binary values. This paper builds three models to analyze the factors that influence family caregiving for disabled elderly in a comprehensive way. Model 1 is based on pre-dispositional factors, and models 2 and 3 build on model 1 by incorporating demand and enabling factors one at a time. The specific models that are built are as follows:

Model 1: $f^1(p) = \alpha^1 + \beta_1^1 * X$ predisposing factors $+ \varepsilon_i^1$

Model 2: $f^2(p) = \alpha^2 + \beta_1^2 * X$ predisposing factors $+ \beta_2^2 * X$ enabling factors $+ \varepsilon_i^2$

Model 3:

$f^3(p) = \alpha^3 + \beta_1^3 * X$ predisposing factors $+ \beta_2^3 * X$ enabling factors $+ \beta_3^3 * X$ need factors $+ \varepsilon_i^3$

With a significance level of $\alpha = 0.05$, CLHLS 2018 data were processed and analyzed using Stata 16.0 and SPSS 25.0.

### Results

Table 1 displays the descriptive statistics for the fundamental variables. In all, 1,318 disabled elderly participated in this study. The majority of family caregivers—87.4%—were the primary caregivers for their children; women chose family care at a rate significantly higher than that of men—68.1%; over 80% of them lived in townships and cities; a significant proportion of them were single; additionally, a higher percentage of disabled elderly—93.5% of all disabled elderly—chosen family care. Additionally, the number of elderly people with mild disabilities choosing family care is higher than the number of elderly people with moderate disabilities and severely disabled people, with 45.3%, 23.7%, and 31.0%, respectively, according to the ADL disability scale. Additionally, more than half of disabled elderly with average self-assessed health preferred family care over those with good and bad self-assessed health.

**Table 1. Univariate analysis of the sample's attributes and the variables affecting family caregiving (%).**

| Variables | Spousal care | Care of children | $\chi^2$ | P |
|---|---|---|---|---|
| **Predisposing factors** | | | | |
| **Age** | | | 273.180 | 0.000 |
| 65~79 | 60(4.6) | 26(2.0) | | |
| 80~ | 106(8.0) | 1126(85.4) | | |
| **Gender** | | | 88.974 | 0.000 |
| Male | 106(8.0) | 315(23.9) | | |
| Female | 60(4.6) | 837(63.5) | | |
| **Education** | | | 60.558 | 0.000 |
| Educated | 81(6.1) | 242(18.4) | | |
| Not educated | 85(6.4) | 910(69.0) | | |
| **Marital status** | | | 740.573 | 0.000 |
| Couple | 6(0.5) | 1062(80.6) | | |
| Single | 160(12.1) | 90(6.8) | | |
| **Enabling factors** | | | | |
| **Residence** | | | 2.716 | 0.099 |
| Rural | 939(71.2) | 144(10.9) | | |
| Urban | 213(16.2) | 22(1.7) | | |
| **Property ownership** | | | 138.870 | 0.000 |
| Yes | 105(8.0) | 236(17.9) | | |
| No | 61(4.6) | 916(69.5) | | |
| **Economic status** | | | 5.082 | 0.079 |
| Rich | 20(1.5) | 193(14.6) | | |
| Average | 112(8.5) | 789(59.9) | | |
| Poor | 34(2.6) | 170(12.9) | | |
| **Old-age insurance** | | | 2.589 | 0.108 |
| Yes | 69(5.2) | 97(7.4) | | |
| No | 405(30.7) | 747(56.7) | | |
| **Community living care** | | | 0.318 | 0.573 |
| Yes | 5(0.4) | 45(3.4) | | |
| No | 161(12.2) | 1107(84.0) | | |
| **Community medical service** | | | 2.981 | 0.084 |
| Yes | 48(3.6) | 263(20.0) | | |
| No | 118(9.0) | 889(67.5) | | |
| **Willingness of primary caregiver** | | | 0.183 | 0.669 |
| Yes | 164(12.4) | 1133(86.0) | | |
| Others | 2(0.2) | 19(1.4) | | |
| **Need factors** | | | | |
| **Self-assessed health** | | | 62.505 | 0.000 |
| Bad | 68(5.2) | 168(13.5) | | |
| Average | 66(5.0) | 625(47.4) | | |
| Good | 32(2.4) | 349(26.5) | | |
| **Degree of disability** | | | 11.180 | 0.004 |
| Mild disability | 89(6.8) | 507(38.5) | | |
| Moderate disability | 44(3.3) | 269(20.4) | | |
| Severe disability | 33(2.5) | 376(28.5) | | |

Among the predisposing factors in Model 1, age, gender, and marital status were associated with home care. The regression results for age showed that in family care, lower-aged disabled elderly (OR = 3.329, $P<0.01$) were more likely to receive spousal care, and higher-aged disabled elderly (OR = 0.300, $P<0.01$) were relatively more likely to receive child care. The regression results for gender showed that male disabled elderly (OR = 0.510, $P < 0.05$) were more likely to receive spousal care, while female (OR = 1.962, $P < 0.05$) disabled elderly were more likely to be cared for by their children; disabled elderly with spouses (OR = 0.005, $P < 0.01$) would receive more care from their spouses, while divorced, widowed or unmarried (OR = 207.002, $P < 0.01$) disabled elderly receive more family care from their children.

With the inclusion of enabling factors in Model 2, the significance of the effect of age was weakened, and more significant was the attribution of housing ownership, number of children, and primary caregiver willingness. The regression results reflect that the more the number of children (OR = 1.232, $P < 0.01$), the more likely the disabled elderly are to receive care from their children; the exchange of housing title belonging (OR = 1.774, $P < 0.05$) promotes the provision of care by children and reduces the likelihood of spousal care; and the more positive the willingness to care of the primary caregiver (OR = 6.536, $P < 0.05$), the more the disabled elderly are likely to receive spousal care.

Predisposing, enabling, and need factors are combined in Model 3. The findings demonstrated that self-assessed health had a major impact on family caring among the demand components. The results of the study are shown in Table 2.

## Discussion

The study's findings indicate that, in the context of aging, the majority of disabled elderly who select family care are 80 years of age or older, and they are primarily found in rural locations. Family care is an important part of the long-term care system for the elderly. It is not only in line with China's traditional long-held concept of filial piety, but it is also used to maintain family emotional ties. For a long time, family care has been the primary choice of care for disabled elderly. It is advised that in response to the circumstances at hand, government agencies define the family as the fundamental unit of care and welfare. As the primary caregivers, children and their spouses must understand the role of the family. They must also raise their children to be contributing members of the family and recognize that, in the event that an elderly person becomes disabled, family members have an unwavering duty to care for them [26]. On the other hand, the government bears the primary responsibility for guaranteeing the smooth functioning of the long-term care system. As the primary source of support, it must ensure that the administrative duties are carried out in a way that complements and cooperates with the family care obligation [27].

Furthermore, the prior study found that a number of enabling characteristics, including housing ownership, the number of children, and the primary caregiver's willingness, all had an impact on the family's ability to care for disabled elderly. According to existing research, the more children there are and the larger the family size, the more the disabled elderly choose family care. This reflects the reality that lower family sizes reduce family caregiver capability [28]. In China, family caregivers for disabled elderly provide varying degrees of care, and providing long-term care for disabled elderly might be risky. In order to put the family care welfare system into action, make family care for disabled elderly a national policy, and create a number of support policies for family care and its primary caregivers, it is imperative that laws be strengthened. In order to support spousal family caregivers, it is advised to maximize the path of financial subsidies for families of disabled elderly and to directly exempt elderly families from paying taxes. In addition, training in caregiving knowledge and skills should be

**Table 2. Binary logistic regression analyses of family care for disabled elderly.**

| Variables | Spousal care | | | Care of children | | |
|---|---|---|---|---|---|---|
| | Model 1 Exp(B) | Model 2 Exp(B) | Model 3 Exp(B) | Model 1 Exp(B) | Model 2 Exp(B) | Model 3 Exp(B) |
| **Age (80~)** | | | | | | |
| 65~79 | 3.329*** | 2.461** | 2.308** | 0.300*** | 0.406** | 0.433** |
| **Gender(Male)** | | | | | | |
| Female | 0.510** | 0.480** | 0.420*** | 1.962** | 2.083** | 2.383*** |
| **Education(Educated)** | | | | | | |
| Not educated | 1.347 | 1.248 | 1.446 | 0.743 | 0.801 | 0.692 |
| **Marital status(Couple)** | | | | | | |
| Single | 0.005*** | 0.005*** | 0.005*** | 207.002*** | 195.893*** | 194.644*** |
| **Residence(Urban)** | | | | | | |
| Rural | | 1.558 | 1.574 | | 0.642 | 0.635 |
| **Property ownership (Yes)** | | | | | | |
| No | | 0.564** | 0.618* | | 1.774** | 1.619* |
| **Economic status (Poor)** | | | | | | |
| Rich | | 0.620 | 0.758 | | 1.613 | 1.319 |
| Average | | 0.875 | 1.040 | | 1.143 | 0.962 |
| **No. of children** | | 0.812*** | 0.801*** | | 1.232*** | 1.249*** |
| **Old-age insurance (Yes)** | | | | | | |
| No | | 0.922 | 0.891 | | 1.019 | 1.122 |
| **Community living care (Yes)** | | | | | | |
| No | | 2.930 | 2.322 | | 0.341 | 0.431 |
| **Community medical service(Yes)** | | | | | | |
| No | | 0.803 | 0.854 | | 1.245 | 1.171 |
| **Willingness of primary caregiver(Others)** | | | | | | |
| Yes | | 6.536** | 10.220** | | 0.153** | 0.098** |
| **Self-assessed health (Bad)** | | | | | | |
| Average | | | 0.298*** | | | 3.357*** |
| Good | | | 0.293*** | | | 3.409*** |
| **Degree of disability (Severe)** | | | | | | |
| Mild disability | | | 1.652 | | | 0.605 |
| Moderate disability | | | 1.372 | | | 0.729 |
| Constant | 1.438 | 0.311 | 0.363 | 0.696 | 3.211 | 2.753 |

*$p<0.1$;

**$p<0.05$;

***$p<0.01$

provided, a social environment of "active ageing" should be fostered, and the social value of elderly spouses providing care should be affirmed [21, 29]. Support for family caregivers of children focuses on guiding multi-party participation, promoting social insurance departments, community organizations, social voluntary organizations, and so on to provide multi-party support to family caregivers; it focuses on strengthening the work balance policy, developing a family-friendly employment system for care for disabled elderly in enterprises and public institutions, and promoting the system of paid nursing care leave to make it more accessible to family caregivers [30].

Similarly, the subjective needs of disabled elderly influence home care, which is consistent with previous findings that both the degree of self-care and self-assessed health of older people influence the choice of care model [5, 31, 32], implying that the home care needs of disabled elderly are fluid and that self-perceived health may differ from reality. Traditional long-term care evaluation instruments are biased in capturing the actual needs of the elderly with impairments as society and medical norms advance. The ADL disability measurement framework is widely recognized and is still in use today, however it lacks assessment of the elderly's mental health and social interaction, which is not conducive to scientific judgment of actual care needs [5]. Given the aforementioned analyses, it is advised to establish a system for determining and evaluating the needs of the elderly who are disabled for family care. Additionally, it is advised to draw on the experience of other developed nations to develop unified standards and procedures for evaluating the needs that are appropriate for China's national circumstances [33]. These should take into account the elderly who are disabled's physical and mental well-being, as well as their capacity for communication and interaction with others, as well as other aspects of the broader conception of health [25]. In order to support disabled elderly in accurately evaluating their own health status and to give society a foundation upon which to appropriately assess the long-term care that these individuals require. Following the needs assessment, more work should be put into building follow-up care infrastructure services to support families in achieving the goal of family care and to lessen the burden on them. Additionally, community-based primary health care service stations should be funded based on the current geographic boundaries of the community. Family social work ought to be based on the disabled family as a unit, encouraging the development of an integrated family-community care model and offering "daytime respite" services to family caregivers as a means of temporary community care.

## Limitations and future directions

The following are some of the limitations of this study: first, it concentrates on the spouses and children of disabled elderly and ignores other relatives who might offer family care, which could introduce confounding bias into the data. Secondly, the cross-sectional study did not investigate causation; therefore, longitudinal research is also required to confirm that family caring is positively correlated with the influencing factors. Furthermore, while CLHLS covers two-thirds of China's provinces and represents the basic situation of the elderly in China, this database is not specific to the group of disabled elderly, and information about them is limited, making it impossible to measure the influencing factors of family care comprehensively. Further longitudinal research is required to expand the study's scope and synthesize the potential contributing components.

## Conclusion

Children have a significant role in family care. Predisposing factors included younger male disabled elderly receiving spousal care and female disabled elderly without husbands receiving care from their offspring. Among the enabling factors, the number of children is positively related to the likelihood of receiving care from children; housing property ownership attribution has a positive incentive effect on children's caregiving behavior; and, according to the theory of intergenerational exchange, children can receive a gift of property from the elderly by providing care and assistance to disabled elderly [34]. Disabled elderly are more likely to choose spousal care if their spouse is willing to take on the role of primary caregiver. One need factor that influences family caregiving is disabled elderly's health self-assessment; those with positive results are more likely to have their children provide care, while those with negative

results are more likely to have their spouses provide it. To summarize, the government should provide more flexible and diverse assistance policies for family caregivers of the disabled elderly. It should not only enhance related welfare policies and expedite the development of a long-term care insurance system to provide financial security for family caregivers, but also develop new forms of community care to adequately share the burden of care for family caregivers. Furthermore, a mechanism for recognizing and grading the care needs of families of disabled elderly should be developed in order to reliably identify care needs and offer a scientific foundation for continual improvement of the long-term care services and systems.

## Author Contributions

**Data curation:** Li Wang.

**Funding acquisition:** Li Wang.

**Methodology:** Yuwei Zhang.

**Software:** Yuwei Zhang.

**Writing – original draft:** Yuwei Zhang.

**Writing – review & editing:** Li Wang.

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
