## [Decision Letter · Decision Letter 0]

27 Aug 2024

PONE-D-24-18004Family care and predictors of the disabled elderly in China: a cross-sectional study based on the Anderson modelPLOS ONE

Dear Dr. wang,

Thank you for submitting your manuscript to PLOS ONE. After careful consideration, we feel that it has merit but does not fully meet PLOS ONE’s publication criteria as it currently stands. Therefore, we invite you to submit a revised version of the manuscript that addresses the points raised during the review process.

We look forward to receiving your revised manuscript.

Kind regards,

Robbert Huijsman, PhD

Academic Editor

PLOS ONE

Journal Requirements:

   " This study was funded by a Study on Guangzhou's Active Response to Population Ageing: Identification of Dilemmas and Breakthroughs in Guangzhou's Smart Health and Aging Services under the Ecosystem View(2022GZYB37),A study on the holistic governance of the multiple supply of long-term care services in China(20FGLB048),Key Laboratory of Philosophy and Social Sciences of Guangdong Higher Education Institutions for Health Polices Research and Evaluation (2015WSY0010)."

4.  Please note that your Data Availability Statement is currently missing the repository name. If your manuscript is accepted for publication, you will be asked to provide these details on a very short timeline. We therefore suggest that you provide this information now, though we will not hold up the peer review process if you are unable.

Additional Editor Comments:

We thank the authors for their paper. We apologize for the long review period, as we experienced difficulties to find reviewers. With one reviewer report available we have come to our decision to ask for major revision.

Please use the good comments of the reviewer to revise your paper. Main issues are the selection of "predictors" for family care, with very much literature available (especially burden of care, further discrimination to different care tasks?) and the embedding of your discussion in both your own results (which should substantiate your conclusions) and literature.

Reviewers' comments:

Reviewer's Responses to Questions

**Comments to the Author**

1. Is the manuscript technically sound, and do the data support the conclusions?

Reviewer #1: Partly

2. Has the statistical analysis been performed appropriately and rigorously? 

Reviewer #1: Yes

3. Have the authors made all data underlying the findings in their manuscript fully available?

Reviewer #1: Yes

4. Is the manuscript presented in an intelligible fashion and written in standard English?

Reviewer #1: No

5. Review Comments to the Author

Reviewer #1: The main problem of this study is that the results are very much about who is providing family care (spouses and children). No information is given on the burden that family members might experience in providing care. In the discussion it is mentioned "family care is difficult ..." and that family member are at risk of mental and physical health detoriation. However, there are no results that substantiates these conclusions.

The abstract is not always clear

line 25: aging and aging .. what does this mean?

line 38-40: the conclusion is not supported by the data

Introduction:

line 18, 19: this gives the impression that elderly disabled have to rely on informal care since formal care cannot meet the demands.

line 59: 60+? 65+

line 61: however ?

line 68-70: sustainability ... is under threat. If this is the heart of the research than other data needed to be accumulated, i.e. on the burden of informal care

Results:

146: slightly?

149: average? in table 1: moderate ?

150: poor: in table 1 mild ?

Discussion:

176, 177: Family care is difficult: there are no data presented here to support this

180: put spouses and children at risk: no data presented in this research to support this claim

198, 199: specific assistance is required: no data to support this

6. PLOS authors have the option to publish the peer review history of their article (what does this mean?). If published, this will include your full peer review and any attached files.

Reviewer #1: **Yes: **dr. Harry Finkenflugel

---

## [Author Response · Author response to Decision Letter 0]

25 Sep 2024

Response Letter

Dear editors and reviewers,

We are very grateful for your constructive comments and suggestions for our manuscript entitled" Family care and predictors of the disabled elderly in China: a cross-sectional study based on the Anderson model"(lD:PONE-D-24-18004).Your comments are very valuable and helpful for improving our manuscript. In the following the responses to all the comments are provided one by one.We have tried our best to make all there visions clear, and we hope that the revised manuscript can satisfy the requirements for publication.The main revisions in the new manuscript are:

1. More specific expressions have been used;

2. Abstract has been updated;

3. Introduction has been supplemented.

Sincerely,

Corresponding author.

Response to the comments of Reviewer #1

Q1. The manuscript was not presented in an easily understandable manner and written in standard English.

Thank you for your insightful question, line 25: aging and aging has been changed to “In light of China's progressively aging population” , line 146: slightly,line149: average, in table 1: moderate,line150: poor: in table 1 mild have been changed to “Mild disability”、“Moderate disability”、“Severe disability”

Q2. The main problem of this study is that the results are very much about who is providing family care (spouses and children). No information is given on the burden that family members might experience in providing care. In the discussion it is mentioned "family care is difficult ..." and that family member are at risk of mental and physical health detoriation. However, there are no results that substantiates these conclusions.

Thank you for your insightful comment and kind suggestion. Family members might experience in providing care has been supplemented. "family care is difficult ..."has been deleted.

Q3. The abstract is not always clear. The abstract is not always clear line 25: aging and aging .. what does this mean?line 38-40: the conclusion is not supported by the data.

Thank you very much for the constructive feedback.The conclusions have been modified in the light of the results of the study.

Q4.Introduction: line 18, 19: this gives the impression that elderly disabled have to rely on informal care since formal care cannot meet the demands.line 59: 60+? 65+.line 61: however ?.line 68-70: sustainability ... is under threat. If this is the heart of the research than other data needed to be accumulated, i.e. on the burden of informal care

Thank you for your remainder.The expression of ambiguity has been clarified, more data about China's elderly population has been included, and a body of literature has been gathered regarding the variables affecting the caregiving burden on families.

Q5. Discussion:176, 177: Family care is difficult: there are no data presented here to support this.180: put spouses and children at risk: no data presented in this research to support this claim.198, 199: specific assistance is required: no data to support this.

We thank the reviewer for the highly valuable comment. Discussions that lacked evidence have been removed.

Response to the comments of Additional Editor

Q1. Main issues are the selection of "predictors" for family care, with very much literature available (especially burden of care, further discrimination to different care tasks?) and the embedding of your discussion in both your own results (which should substantiate your conclusions) and literature.

We thank the reviewer for the highly valuable comment.The relevant available literature has been added and the discussion will be embedded in its own results.

The paper has been revised carefully and thoroughly according to the comments from the reviewers. All the questions from the reviewers have been answered.Thanks to the professional comments again that point out the above problems. The authors hope these explanations would answer your doubts.

---

## [Editor Report · Decision Letter 1]

30 Sep 2024

Family care and predictors of the disabled elderly in China: a cross-sectional study based on the Anderson model

PONE-D-24-18004R1

Dear Dr. wang,

We’re pleased to inform you that your manuscript has been judged scientifically suitable for publication and will be formally accepted for publication once it meets all outstanding technical requirements.

Kind regards,

Robbert Huijsman, PhD

Academic Editor

PLOS ONE

Additional Editor Comments (optional):

Dear Authors, Thanks for your revision. It answers the reviewers' comments adequately.
---

## [Editor Report · Acceptance letter]

24 Oct 2024

PONE-D-24-18004R1 

PLOS ONE

Dear Dr. wang, 

I'm pleased to inform you that your manuscript has been deemed suitable for publication in PLOS ONE. Congratulations! Your manuscript is now being handed over to our production team.

Kind regards, 

on behalf of

Professor Robbert Huijsman 

Academic Editor

PLOS ONE